# Correlation in Extensive-Form Games: Saddle-Point Formulation and Benchmarks[*]

**Gabriele Farina**
Computer Science Department
Carnegie Mellon University
gfarina@cs.cmu.edu

**Chun Kai Ling**
Computer Science Department
Carnegie Mellon University
chunkail@cs.cmu.edu

**Fei Fang**
Institute for Software Research
Carnegie Mellon University
feif@cs.cmu.edu

**Tuomas Sandholm**
Computer Science Department, CMU
Strategic Machine, Inc.
Strategy Robot, Inc.
Optimized Markets, Inc.
sandholm@cs.cmu.edu

## Abstract

While Nash equilibrium in extensive-form games is well understood, very little is known about the properties of *extensive-form correlated equilibrium (EFCE)*, both from a behavioral and from a computational point of view. In this setting, the strategic behavior of players is complemented by an external device that privately recommends moves to agents as the game progresses; players are free to deviate at any time, but will then not receive future recommendations. Our contributions are threefold. First, we show that an EFCE can be formulated as the solution to a bilinear saddle-point problem. To showcase how this novel formulation can inspire new algorithms to compute EFCEs, we propose a simple subgradient descent method which exploits this formulation and structural properties of EFCEs. Our method has better scalability than the prior approach based on linear programming. Second, we propose two benchmark games, which we hope will serve as the basis for future evaluation of EFCE solvers. These games were chosen so as to cover two natural application domains for EFCE: conflict resolution via a mediator, and bargaining and negotiation. Third, we document the qualitative behavior of EFCE in our proposed games. We show that the social-welfare-maximizing equilibria in these games are highly nontrivial and exhibit surprisingly subtle *sequential* behavior that so far has not received attention in the literature.

## 1 Introduction

Nash equilibrium (NE) [Nash, 1950], the most seminal concept in non-cooperative game theory, captures a multi-agent setting where each agent is selfishly motivated to maximize their own payoff. The assumption underpinning NE is that the interaction is completely *decentralized*: the behavior of each agent is not regulated by any external orchestrator. Contrasted with the other—often utopian— extreme of a fully managed interaction, where an external dictator controls the behavior of each agent so that the whole system moves to a desired state, the social welfare that can be achieved by NE is generally lower, sometimes dramatically so [Koutsoupias and Papadimitriou, 1999; Roughgarden and Tardos, 2002]. Yet, in many realistic interactions, some intermediate form of centralized control can be achieved. In particular, in his landmark paper, Aumann [1974] proposed the concept of *correlated*

---

[*]The full version of this paper is available on arXiv.

*equilibrium* (CE), where a mediator (the *correlation device*) can *recommend* behavior, but not *enforce it*. In a CE, the correlation device is constructed so that the agents—which are still modeled as fully rational and selfish just like in an NE—have no incentive to deviate from the private recommendation. Allowing correlation of actions while ensuring selfishness makes CE a good candidate solution concept in multi-agent and semi-competitive settings such as traffic control, load balancing [Ashlagi *et al.*, 2008], and carbon abatement [Ray and Gupta, 2009], and it can lead to win-win outcomes.

In this paper, we study the natural extension of correlated equilibrium in *extensive-form* (i.e., sequential) games, known as extensive-form correlated equilibrium (EFCE) [von Stengel and Forges, 2008]. Like CE, EFCE assumes that the strategic interaction is complemented by an external mediator; however, in an EFCE the mediator only privately reveals the recommended next move to each *acting player*, instead of revealing the whole plan of action throughout the game (i.e., recommended move at *all* decision points) for each player at the beginning of the game. Furthermore, while each agent is free to defect from the recommendation at any time, this comes at the cost of future recommendations.

While the properties of correlation in *normal-form* games are well-studied, they do not automatically transfer to the richer world of sequential interactions. It is known in the study of NE that sequential interactions can pose different challenges, especially in settings where the agents retain private information. Conceptually, the players can strategically adjust to dynamic observations about the environment and their opponents as the game progresses. Despite tremendous interest and progress in recent years for computing NE in sequential interactions with private information, with significant milestones achieved in poker games [Bowling *et al.*, 2015; Brown and Sandholm, 2017; Moravčík *et al.*, 2017; Brown and Sandholm, 2019b] and other large, real-world domains, not much has been done to increase our understanding of (extensive-form) correlated equilibria in these settings.

**Contributions**    Our primary objective with this paper is to spark more interest in the community towards a deeper understanding of the behavioral and computational aspects of EFCE.

- In Section 3 we show that an EFCE in a two-player general-sum game is the solution to a bilinear saddle-point problem (BSPP). This conceptual reformulation complements the EFCE construction by von Stengel and Forges [2008], and allows for the development of new and efficient algorithms. As a proof of concept, by using our reformulation we devise a variant of projected subgradient descent which outperforms linear-programming(LP)-based algorithms proposed by von Stengel and Forges [2008] in large game instances.
- In Section 5 we propose two benchmark games; each game is parametric, so that these games can scale in size as desired. The first game is a general-sum variant of the classic war game *Battleship*. The second game is a simplified version of the *Sheriff of Nottingham* board game. These games were chosen so as to cover two natural application domains for EFCE: conflict resolution via a mediator, and bargaining and negotiation.
- By analyzing EFCE in our proposed benchmark games, we show that even if the mediator cannot enforce behavior, it can induce significantly higher social welfare than NE and successfully deter players from deviating in at least two (often connected) ways: (1) using certain sequences of actions as 'passcodes' to verify that a player has not deviated: defecting leads to incomplete or wrong passcodes which indicate deviation, and (2) inducing opponents to play punitive actions against players that have deviated from the recommendation, if such a deviation is detected. Crucially, both deterrents are unique to *sequential* interactions and do not apply to non-sequential games. This corroborates the idea that the mediation of sequential interactions is a qualitatively different problem than that of non-sequential games and further justifies the study of EFCE as an interesting direction for the community. To our knowledge, these are the first experimental results and observations on EFCE in the literature.

The source code for our game generators and subgradient method is published online[2].

## 2    Preliminaries

Extensive-form games (EFGs) are sequential games that are played over a rooted game tree. Each node in the tree belongs to a player and corresponds to a decision point for that player. Outgoing edges from a node $v$ correspond to actions that can be taken by the player to which $v$ belongs. Each terminal node in the game tree is associated with a tuple of payoffs that the players receive should

the game end in that state. To capture imperfect information, the set of vertices of each player is partitioned into *information sets*. The vertices in a same information set are indistinguishable to the player that owns those vertices. For example, in a game of Poker, a player cannot distinguish between certain states that only differ in opponent's private hand. As a result, the strategy of the player (specifying which action to take) is defined on the information sets instead of the vertices. For the purpose of this paper, we only consider *perfect-recall* EFGs. This property means that each player does not forget any of their previous action, nor any private or public observation that the player has made. The perfect-recall property can be formalized by requiring that for any two vertices in a same information set, the paths from those vertices to the root of the game tree contain the exact same sequence of actions for the acting player at the information set.

A pure normal-form strategy for Player $i$ defines a choice of action for *every* information set that belongs to $i$. A player can play a mixed strategy, i.e., sample from a distribution over their pure normal-form strategies. However, this representation contains redundancies: some information sets for Player $i$ may become unreachable reachable after the player makes certain decisions higher up in the tree. Omitting these redundancies leads to the notion of *reduced-normal-form* strategies, which are known to be strategically equivalent to normal-form strategies (e.g., [Shoham and Leyton-Brown, 2009] for more details). Both the normal-form and the reduced-normal-form representation are exponentially large in the size of the game tree.

Here, we fix some notations. Let $Z$ be the set of terminal states (or equivalently, outcomes) in the game and $u_i(z)$ be the utility obtained by player $i$ if the game terminates at $z \in Z$. Let $\Pi_i$ be the set of pure reduced-normal-form strategies for Player $i$. We define $\Pi_i(I)$, $\Pi_i(I, a)$ and $\Pi_i(z)$ to be the set of reduced-normal-form strategies that (a) can lead to information set $I$, (b) can lead to $I$ and prescribes action $a$ at information set $I$, and (c) can lead to the terminal state $z$, respectively. We denote by $\Sigma_i$ the set of information set-action pairs $(I, a)$ (also referred to as *sequences*), where $I$ is an information set for Player $i$ and $a$ is an action at set $I$. For a given terminal state $z$ let $\sigma_i(z)$ be the last $(I, a)$ pair belonging to Player $i$ encountered in the path from the root of the tree to $z$.

**Extensive-Form Correlated Equilibrium** Extensive-form correlated equilibrium (EFCE) is a solution concept for extensive-form games introduced by von Stengel and Forges [2008].[3] Like in the traditional correlated equilibrium (CE), introduced by Aumann [1974], a *correlation device* selects private signals for the players before the game starts. These signals are sampled from a correlated distribution $\mu$—a joint probability distribution over $\Pi_1 \times \Pi_2$—and represent recommended player strategies. However, while in a CE the recommended moves for the whole game tree are privately revealed to the players when the game starts, in an EFCE the recommendations are revealed incrementally as the players progress in the game tree. In particular, a recommended move is only revealed when the player reaches the decision point in the game for which the recommendation is relevant. Moreover, if a player ever deviates from the recommended move, they will stop receiving recommendations. To concretely implement an EFCE, one places recommendations into 'sealed envelopes' which may only be opened at its respective information set. Sealed envelopes may implemented using cryptographic techniques (see Dodis *et al.* [2000] for one such example).

In an EFCE, the players know less about the set of recommendations that were sampled by the correlation device. The benefits are twofold. First, the players can be more easily induced to play strategies that hurt them (but benefit the overall social welfare), as long as "on average" the players are indifferent as to whether or not to follow the recommendations: the set of EFCEs is a *superset* of that of CEs. Second, since the players observe less, the set of probability distributions for the correlation device for which no player has an incentive to deviate can be described succinctly in certain classes of games: von Stengel and Forges [2008, Theorem 1.1] show that in two-player, perfect-recall extensive-form games with no chance moves, the set of EFCEs can be described by a system of linear equations and inequalities of polynomial size in the game description. On the other hand, the same result cannot hold in more general settings: von Stengel and Forges [2008, Section 3.7] also show that in games with more than two players and/or chance moves, deciding the existence of an EFCE with social welfare greater than a given value is NP-hard. It is important to note that this last result only implies that the characterization of the set of *all* EFCEs cannot be of polynomial size in general (unless P = NP). However, the problem of finding *one* EFCE can be

solved in polynomial time: Huang [2011] and Huang and von Stengel [2008] show how to adapt the *Ellipsoid Against Hope* algorithm [Papadimitriou and Roughgarden, 2008; Jiang and Leyton-Brown, 2015] to compute an EFCE in polynomial time in games with more than two players and/or with chance moves. Unfortunately, that algorithm is only theoretical, and known to not scale beyond extremely small instances [Leyton-Brown, 2019].

## 3 Extensive-Form Correlated Equilibria as Bilinear Saddle-Point Problems

Our objective for this section is to cast the problem of finding an EFCE in a two-player game as a bilinear saddle-point problem, that is a problem of the form $\min_{x \in \mathcal{X}} \max_{y \in \mathcal{Y}} \ x^\top A y$, where $\mathcal{X}$ and $\mathcal{Y}$ are compact convex sets. In the case of EFCE, $\mathcal{X}$ and $\mathcal{Y}$ are convex polytopes that belong to a space whose dimension is polynomial in the game tree size. This reformulation is meaningful:

- From a conceptual angle, it brings the problem of computing an EFCE closer to several other solution concepts in game theory that are known to be expressible as BSPP. In particular, the BSPP formulation shows that an EFCE can be viewed as a NE in a two-player zero-sum game between a *deviator*, who is trying to decide how to best defect from recommendations, and a *mediator*, who is trying to come up with an incentive-compatible set of recommendations.
- From a geometric point of view, the BSPP formulation better captures the combinatorial structure of the problem: $\mathcal{X}$ and $\mathcal{Y}$ have a well-defined meaning in terms of the input game tree. This has algorithmic implications: for example, because of the structure of $\mathcal{Y}$ (which will be detailed later), the inner maximization problem can be solved via a single bottom-up game-tree traversal.
- From a computational standpoint, it opens the way to the plethora of optimization algorithms (both general-purpose and those specific to game theory) that have been developed to solve BSPPs. Examples include Nesterov's excessive gap technique [Nesterov, 2005], Nemirovski's mirror prox algorithm [Nemirovski, 2004] and regret-methods based methods such as mirror descent, follow-the-regularized-leader (e.g., Hazan [2016]), and CFR and its variants Zinkevich *et al.* [2007]; Farina *et al.* [2019]; Brown and Sandholm [2019a].

Furthermore, it is easy to show that by dualizing the inner maximization problem in the BSPP formulation, one recovers the linear program introduced by von Stengel and Forges [2008] (we show this in Appendix A in the full paper). In this sense, our formulation subsumes the existing one.

**Triggers and Deviations** One effective way to reason about extensive-form correlated equilibria is via the notion of *trigger agents*, which was introduced (albeit used in a different context) in Gordon *et al.* [2008] and Dudik and Gordon [2009]:

**Definition 1.** *Let $\hat{\sigma} := (\hat{I}, \hat{a}) \in \Sigma_i$ be a sequence for Player $i$, and let $\hat{\mu}$ be a distribution over $\Pi_i(\hat{I})$. A $(\hat{\sigma}, \hat{\mu})$-trigger agent for Player $i$ is a player that follows all recommendations given by the mediator unless they get recommended $\hat{a}$ at $\hat{I}$; in that case, the player 'gets triggered', stops following the recommendations and instead plays based on a pure strategy sampled from $\hat{\mu}$ until the game ends.*

A correlated distribution $\mu$ is an EFCE if and only if any trigger agent for Player $i$ can get utility at most equal to the utility that Player $i$ earns by following the recommendations of the mediator at all decision points. In order to express the utility of the trigger agent, it is necessary to compute the probability of the game ending in each of the terminal states. As we show in Appendix B in the full paper, this can be done concisely by partitioning the set of terminal nodes in the game tree into three different sets. In particular, let $Z_{\hat{I}, \hat{a}}$ be the set of terminal nodes whose path from the root of the tree contains taking action $\hat{a}$ at $\hat{I}$ and let $Z_{\hat{I}}$ be the set of terminal nodes whose path from the root passes through $\hat{I}$ and are *not* in $Z_{\hat{I}, \hat{a}}$. We have

**Lemma 1.** *Consider a $(\hat{\sigma}, \hat{\mu})$-trigger agent for Player 1, where $\hat{\sigma} = (\hat{I}, \hat{a})$. The value of the trigger agent, defined as the expected difference between the utility of the trigger agent and the utility of an agent that always follows recommendations sampled from correlated distribution $\mu$, is computed as*

$$v_{1,\hat{\sigma}}(\mu, \hat{\mu}) := \sum_{z \in Z_{\hat{I}}} u_1(z) \xi_1(\hat{\sigma}; z) y_{1,\hat{\sigma}}(z) - \sum_{z \in Z_{\hat{I}, \hat{a}}} u_1(z) \xi_1(\sigma_1(z); z),$$

*where $\xi_1(\hat{\sigma}; z) := \sum_{\pi_1 \in \Pi_1(\hat{\sigma})} \sum_{\pi_2 \in \Pi_2(z)} \mu(\pi_1, \pi_2)$ and $y_{1,\hat{\sigma}}(z) := \sum_{\hat{\pi}_1 \in \Pi_1(z)} \hat{\mu}(\hat{\pi}_1)$.*

(A symmetric result holds for Player 2, with symbols $\xi_2(\hat{\sigma}; z)$ and $y_{2,\hat{\sigma}}(z)$.) It now seems natural to perform a change of variables, and pick distributions for the random variables $y_{1,\hat{\sigma}}(\cdot), y_{2,\hat{\sigma}}(\cdot), \xi_1(\cdot; \cdot)$

and $\xi_2(\cdot;\cdot)$ instead of $\mu$ and $\hat{\mu}$. Since there are only a polynomial number (in the game tree size) of combinations of arguments for these new random variables, this approach allows one to remove the redundancy of realization-equivalent normal-form plans and focus on a significantly smaller search space. In fact, the definition of $\xi = (\xi_1, \xi_2)$ also appears in [von Stengel and Forges, 2008], referred to as (sequence-form) *correlation plan*. In the case of the $y_{1,\hat{\sigma}}$ and $y_{2,\hat{\sigma}}$ random variables, it is clear that the change of variables is possible via the sequence form [von Stengel, 2002]; we let $Y_{i,\hat{\sigma}}$ be the sequence-form polytope of feasible values for the vector $y_{i,\hat{\sigma}}$. Hence, the only hurdle is characterizing the space spanned by $\xi_1$ and $\xi_2$ as $\mu$ varies across the probability simplex. In two-player perfect-recall games with no chance moves, this is exactly one of the merits of the landmark work by von Stengel and Forges [2008]. In particular, the authors prove that in those games the space of feasible $\xi$ can be captured by a polynomial number of linear constraints. In more general cases the same does not hold (see second half of Section 2), but we prove the following (Appendix C in the full paper):

**Lemma 2.** *In a two-player game, as $\mu$ varies over the probability simplex, the joint vector of $\xi_1(\cdot;\cdot)$, $\xi_2(\cdot;\cdot)$ variables spans a convex polytope $\mathcal{X}$ in $\mathbb{R}^n$, where $n$ is at most quadratic in the game size.*

**Saddle-Point Reformulation**   According to Lemma 1, for each Player $i$ and $(\hat{\sigma}, \hat{\mu})$-trigger agent for them, the value of the trigger agent is a biaffine expression in the vectors $y_{i,\hat{\sigma}}$ and $\xi_i$, and can be written as $v_{i,\hat{\sigma}}(\xi_i, y_{i,\hat{\sigma}}) = \xi_i^\top A_{i,\hat{\sigma}} y_{i,\hat{\sigma}} - b_{i,\hat{\sigma}}^\top \xi_i$ for a suitable matrix $A_{i,\hat{\sigma}}$ and vector $b_{i,\hat{\sigma}}$, where the two terms in the difference correspond to the expected utility for deviating at $\hat{\sigma}$ according to the (sequence-form) strategy $y_{i,\hat{\sigma}}$ and the expected utility for not deviating at $\hat{\sigma}$. Given a correlation plan $\xi = (\xi_1, \xi_2) \in \mathcal{X}$, the maximum value of any deviation for any player can therefore be expressed as

$$v^*(\xi) := \max_{\{i,\hat{\sigma},y_{i,\hat{\sigma}}\}} v_{i,\hat{\sigma}}(\xi_i, y_{i,\hat{\sigma}}) = \max_{i\in\{1,2\}} \max_{\hat{\sigma}\in\Sigma_i} \max_{y_{\hat{\sigma}}\in Y_{\hat{\sigma}}} \{\xi_i^\top A_{i,\hat{\sigma}} y_{i,\hat{\sigma}} - b_{i,\hat{\sigma}}^\top \xi_i\}.$$

We can convert the maximization above into a continuous linear optimization problem by introducing the multipliers $\lambda_{i,\hat{\sigma}} \in [0,1]$ (one per each Player $i \in \{1,2\}$ and trigger $\hat{\sigma} \in \Sigma_i$), and write

$$v^*(\xi) = \max_{\{\lambda_{i,\hat{\sigma}}, z_{i,\hat{\sigma}}\}} \sum_i \sum_{\hat{\sigma}} \xi_i^\top A_{i,\hat{\sigma}} z_{i,\hat{\sigma}} - \lambda_{i,\hat{\sigma}} b_{i,\hat{\sigma}}^\top \xi_i,$$

where the maximization is subject to the linear constraints $[C_1] \sum_{i\in\{1,2\}} \sum_{\hat{\sigma}\in\Sigma_i} \lambda_{i,\hat{\sigma}} = 1$ and $[C_2]$ $z_{i,\hat{\sigma}} \in \lambda_{i,\hat{\sigma}} Y_{i,\hat{\sigma}}$ for all $i \in \{1,2\}, \hat{\sigma} \in \Sigma_i$. These linear constraints define a polytope $\mathcal{Y}$.

A correlation plan $\xi$ is an EFCE if an only if $v_{i,\hat{\sigma}}(\xi, y_{i,\hat{\sigma}}) \le 0$ for every trigger agent, i.e., $v^*(\xi) \le 0$. Therefore, to find an EFCE, we can solve the optimization problem $\min_{\xi\in\mathcal{X}} v^*(\xi)$, which is a bilinear saddle point problem over the convex domains $\mathcal{X}$ and $\mathcal{Y}$, both of which are convex polytopes that belong to $\mathbb{R}^n$, where $n$ is at most quadratic in the input game size (Lemma 2). If an EFCE exists, the optimal value should be non-positive and the optimal solution is an EFCE (as it satisfies $v^*(\xi) \le 0$). In fact, since EFCE's always exist (as EFCEs are supersets of CEs von Stengel and Forges [2008]), and one can select triggers to be terminal sequences for Player 1, the optimal value of the BSPP is always 0. The BSPP can be interpreted as the NE of a zero-sum game between the *mediator*, who decides on a suitable correlation plan $\xi$ and a *deviator* who selects the $y_{i,\hat{\sigma}}$'s to maximize each $v_{i,\hat{\sigma}}(\xi_i, y_{i,\hat{\sigma}})$. The value of this game is always 0.

Finally, we can enforce a minimum lower bound $\tau$ on the sum of players' utility by introducing an additional variable $\lambda_{\text{sw}} \in [0,1]$ and maximizing the new convex objective

$$v^*_{\text{sw}}(\xi) := \max_{\lambda_{\text{sw}}\in[0,1]} \left\{ (1-\lambda_{\text{sw}}) \cdot v^*(\xi) + \lambda_{\text{sw}} \left[ \tau - \sum_{z\in Z} u_1(z)\xi_1(z;z) - \sum_{z\in Z} u_2(z)\xi_2(z;z) \right] \right\}. \quad (1)$$

## 4   Computing an EFCE using Subgradient Descent

von Stengel and Forges [2008] show that a SW-maximizing EFCE of a two-player game without chance may be expressed as the solution of an LP and solved using generic methods such as the simplex algorithm or interior-point methods. However, this does not scale to large games as these methods require storing and inverting large matrices. Another way of computing SW-maximizing EFCEs was provided by Dudik and Gordon [2009]. However, their algorithm assumes that sampling from correlation plans is possible using the Monte Carlo Markov chain algorithm and does not factor in convergence of the Markov chain. Furthermore, even though their formulation generalizes beyond

our setting of two-player games without chance, our gradient descent method admits more complex objectives. In particular, it allows the mediator to maximize over general concave objectives (in correlation plans) instead of only linear objectives with potentially some regularization. Here, we showcase the benefits of exploiting the combinatorial structure of the BSPP formulation of Section 3 by proposing a simple algorithm based on subgradient descent; in Section 6 we show that this method scales better than commercial state-of-the-art LP solver in large games.

For brevity, we only provide a sketch of our algorithm, which computes a feasible EFCE; the extension to the slightly more complicated objective $v_{\text{sw}}^*(\xi)$ (Equation 1) is straightforward—see Appendix D in the full paper for more details. First, observe that the objective $v^*(\xi)$ is convex since it is the maximum of linear functions of $\xi$. This suggests that we may perform subgradient descent on $v^*$, where the subgradients are given by $\partial/\partial\xi \, v^*(\xi) = A_{i^*,\hat{\sigma}^*} y_{i^*,\hat{\sigma}^*}^* - b_{i,\hat{\sigma}^*}$, where $(i^*, \hat{\sigma}^*, y_{i^*,\hat{\sigma}^*}^*)$ is a triplet which maximizes the objective function $v^*(\xi)$. The computation of such a triplet can be done via a straightforward bottom-up traversal of the game tree. In order to maintain feasibility (that is, $\xi \in \mathcal{X}$), it is necessary to project onto $\mathcal{X}$, which is challenging in practice because we are not aware of any distance-generating function that allows for efficient projection onto this polytope. This is so even in games without chance (where $\xi$ can be expressed by a polynomial number of constraints [von Stengel and Forges, 2008]). Furthermore, iterative methods such as Dykstra's algorithm, add a dramatic overhead to the cost of each iterate.

To overcome this hurdle, we observe that in games with no chance moves, the set $\mathcal{X}$ of correlation plans—as characterized by von Stengel and Forges [2008] via the notion of consistency constraints—can be expressed as the intersection of three sets: (i) $\mathcal{X}_1$, the sets of vectors $\xi$ that only satisfy consistency constraints for Player 1; (ii) $\mathcal{X}_2$, the sets of vectors $\xi$ that only satisfy consistency constraints for Player 2; and (iii) $\mathbb{R}_+^n$, the non-negative orthant. $\mathcal{X}_1$ and $\mathcal{X}_2$ are polytopes defined by equality constraints only. Therefore, an exact projection (in the Euclidean sense) onto $\mathcal{X}_1$ and $\mathcal{X}_2$ can be carried out efficiently by precomputing a suitable factorization the constraint matrices that define $\mathcal{X}_1$ and $\mathcal{X}_2$. In particular, we are able to leverage the specific combinatorial structure of the constraints that form $\mathcal{X}_1$ and $\mathcal{X}_2$ to design an efficient and parallel sparse factorization algorithm (see Appendix D in the full paper for the full details). Furthermore, projection onto the non-negative orthant can be done conveniently, as it just amounts to computing a component-wise maximum between $\xi$ and the zero vector. Since $\mathcal{X} = \mathcal{X}_1 \cap \mathcal{X}_2 \cap \mathbb{R}_+^n$, and since projecting onto $\mathcal{X}_1$, $\mathcal{X}_2$ and $\mathbb{R}_+^n$ individually is easy, we can adopt the recent algorithm proposed by Wang and Bertsekas [2013] designed to handle exactly this situation. In that algorithm, gradient steps are interlaced with projections onto $\mathcal{X}_1$, $\mathcal{X}_2$ and $\mathbb{R}_+^n$ in a cyclical manner. This is similar to projected gradient descent, but instead of projecting onto the intersection of $\mathcal{X}_1$, $\mathcal{X}_2$ and $\mathbb{R}_+^n$ (which we believe to be difficult), we project onto just one of them in round-robin fashion. This simple method was shown to converge by Wang and Bertsekas [2013]. However, no convergence bound is currently known.

# 5  Introducing the First Benchmarks for EFCE

In this section we introduce the first two benchmark games for EFCE. These games are naturally parametric so that they can scale in size as desired and hence used to evaluate different EFCE solvers. In addition, we show that the EFCE in these games are interesting behaviorally: the correlation plan in social-welfare-maximizing EFCE is highly nontrivial and even seemingly counter-intuitive. We believe some of these induced behaviors may prove practical in real-world scenarios and hope our analysis can spark an interest in EFCEs and other equilibria in sequential settings.

## 5.1  Battleship: Conflict Resolution via a Mediator

In this section we introduce our first proposed benchmark game to illustrate the power of correlation in extensive-form games. Our game is a general-sum variant of the classic game *Battleship*. Each player takes turns to secretly place a set of ships $\mathcal{S}$ (of varying sizes and value) on separate grids of size $H \times W$. After placement, players take turns firing at their opponent—ships which have been hit at all the tiles they lie on are considered destroyed. The game continues until either one player has lost all of their ships, or each player has completed $r$ shots. At the end of the game, the payoff of each player is computed as the sum of the values of the opponent's ships that were destroyed, minus $\gamma$ times the value of ships which they lost, where $\gamma \geq 1$ is called the *loss multiplier* of the game. The *social welfare* (SW) of the game is the sum of utilities to all players.

In order to illustrate a few interesting feature of social-welfare-maximizing EFCE in this game, we will focus on the instance of the game with a board of size $3 \times 1$, in which each player commands just 1 ship of value and length 1, there are 2 rounds of shooting per player, and the loss multiplier is $\gamma = 2$. In this game, the social-welfare-maximizing *Nash* equilibrium is such that each player places their ship and shoots uniformly at random. This way, the probability that Player 1 and 2 will end the game by destroying the opponent's ship is $5/9$ and $1/3$ respectively (Player 1 has an advantage since they act first). The probability that both players will end the game with their ships unharmed is a meagre $1/9$. Correspondingly, the maximum SW reached by any NE of the game is $-8/9$.

In the EFCE model, it is possible to induce the players to end the game with a peaceful outcome—that is, no damage to either ship—with probability $5/18$, 2.5 times of the probability in NE, resulting in a much-higher SW of $-13/18$. Before we continue with more details as to how the mediator (correlation device) is able to achieve this result in the case where $\gamma = 2$, we remark that the benefit of EFCE is even higher when the loss multiplier $\gamma$ increases: Figure 1 (left) shows, as a function of $\gamma$, the probability with which Player 1 and 2 terminate the game by sinking their opponent's ship, if they play according to the SW-maximizing EFCE. For all values of $\gamma$, the SW-maximizing NE remains the same while with a mediator, the probability of reaching a peaceful outcome increases as $\gamma$ increases, and asymptotically gets closer to $1/3$ and the gap between the expected utility of the two players vanishes. This is remarkable, considering Player 1's advantage for acting first.

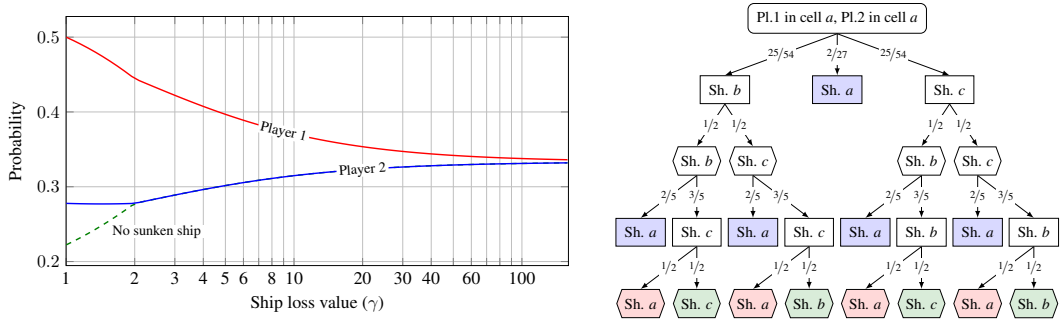

Figure 1: (Left) Probabilities of players sinking their opponent when the players play according to the SW-maximizing EFCE. For $\gamma \geq 2$, the probability of the game ending with no sunken ship and the probability of Player 2 sinking Player 1 coincide. (Right) Example of a playthrough of Battleship assuming both players are recommended to place their ship in the same position $a$. Edge labels represents the probability of an action being recommended. Squares and hexagons denote actions taken by Players 1 and 2 respectively. Blue and red nodes represent cases where Players 1 and 2 sink their opponent, respectively. The *Shoot* action is abbreviated 'Sh.'.

We now resume our analysis of the SW-maximizing EFCE in the instance where $\gamma = 2$. In a nutshell, the correlation plan is constructed in a way that players are recommended to deliberately miss, and deviations from this are punished by the mediator, who reveals to the opponent the ship location that was *recommended* to the deviating player. First, the mediator recommends the players a ship placement that is sampled uniformly at random and independently for each players. This results in 9 possible scenarios (one per possible ship placement) in the game, each occurring with probability $1/9$. Due to the symmetric nature of ship placements, only two scenarios are relevant: whether the two players are recommended to place their ship in the same spot, or in different spots. Figure 1 (right) shows the probability of each recommendation from the mediator in the former case, assuming that the players do not deviate. The latter case is symmetric (see Appendix E in the full paper for details). Now, we explain the first of the two methods in which the mediator compels non-violent behavior. We focus on the first shot made by Player 1 (i.e., the root in Figure 3). The mediator suggests that Player 1 shoot at the Player 2's ship with a low $2/27$ probability, and deliberately miss with high probability. One may wonder how it is possible for this behavior to be incentive-compatible (that is, what are the incentives that compel Player 1 into not defecting), since the player may choose to randomly fire in any of the 2 locations that were *not* recommended, and get almost $1/2$ chance of winning the game immediately. The key is that if Player 1 does so and does not hit the opponent's ship, then the mediator can punish him by recommending that Player 2 shoot in the position where Player 1's was recommended to place their ship. Since players value their ships more than destroying their opponents', the player is incentivized to avoid such a situation by accepting the recommendation to (most probably) miss. We see the first example of deterrent used by the mediator: inducing the opponent to play punitive actions against players that have deviated from the recommendation, if

ever that deviation can be detected from the player. A similar situation arises in the first move of Player 2, where Player 2 is recommended to *deliberately* miss, hitting each of the 2 empty spots with probability $1/2$. A more detailed analysis is available in Appendix E in the full paper.

## 5.2 Sheriff: Bargaining and Negotiation

Our second proposed benchmark is a simplified version of the *Sheriff of Nottingham* board game. The game models the interaction of two players: the *Smuggler*—who is trying to smuggle illegal items in their cargo—and the *Sheriff*—who is trying to stop the Smuggler. At the beginning of the game, the Smuggler secretly loads his cargo with $n \in \{0, \dots, n_{max}\}$ illegal items. At the end of the game, the Sheriff decides whether to inspect the cargo. If the Sheriff chooses to inspect the cargo and finds illegal goods, the Smuggler must pay a fine worth $p \cdot n$ to the Sheriff. On the other hand, the Sheriff has to compensate the Smuggler with a utility $s$ if no illegal goods are found. Finally, if the Sheriff decides not to inspect the cargo, the Smuggler's utility is $v \cdot n$ whereas the Sheriff's utility is $0$. The game is made interesting by two additional elements (which are also present in the board game): *bribery* and *bargaining*. After the Smuggler has loaded the cargo and before the Sheriff chooses whether or not to inspect, they engage in $r$ rounds of bargaining. At each round $i = 1, \dots, r$, the Smuggler tries to tempt the Sheriff into not inspecting the cargo by proposing a bribe $b_i \in \{0, \dots b_{max}\}$, and the Sheriff responds whether or not they would accept the proposed bribe. Only the proposal and response from round $r$ will be executed and have an impact on the final payoffs—that is, all but the $r$-th round of bargaining are non-consequential and their purpose is for the two players to settle on a suitable bribe amount. If the Sheriff accepts bribe $b_r$, then the Smuggler gets $p \cdot n - b_r$, while the Sheriff gets $b_r$. See Appendix F in the full paper for a formal description of the game.

We now point out some interesting behavior of EFCE in this game. We refer to the game instance where $v = 5, p = 1, s = 1, n_{max} = 10, b_{max} = 2, r = 2$ as the *baseline* instance.

**Effect of $v, p$ and $s$.** First, we show what happens in the baseline instance when the item value $v$, item penalty $p$, and Sheriff compensation (penalty) $s$ are varied in isolation over a continuous range of values. The results are shown in Figure 2. In terms of general trends, the effect of the parameter to the Smuggler is fairly consistent with intuition: the Smuggler benefits from a higher item value as well as from higher sheriff penalties, and suffers when the penalty for smuggling is increased. However, the finer details are much more nuanced. For one, the effect of changing the parameters not only is non-monotonic, but also discontinuous. This behavior has never been documented and we find it rather counterintuitive. More counterintuitive observations can be found in Appendix F.
**Effect of $n_{max}, b_{max}$, and $r$.** Here, we try to empirically understand the impact of $n$ and $b$ on the SW

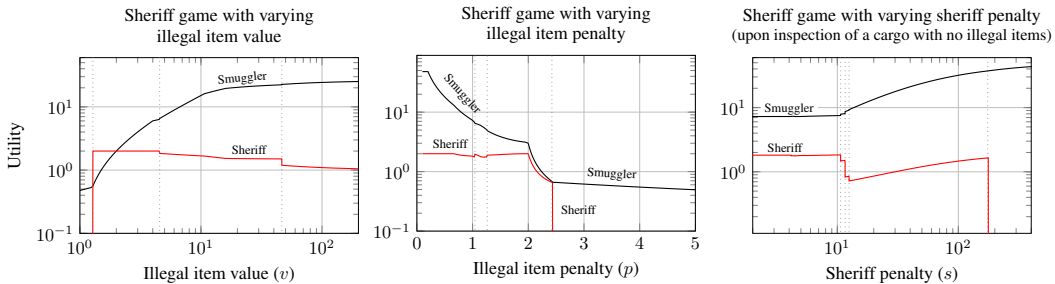

Figure 2: Utility of players with varying $v, p$ and $s$ for the SW-maximizing EFCE. We verified that these plots are not the result of equilibrium selection issues.

maximizing equilibrium. As before we set $v = 5, p = 1, s = 1$ and vary $n$ and $r$ simultaneously while keeping $b_{max}$ constant. The results are shown in Table 1. The most striking observation is that increasing the capacity of the cargo $n_{max}$ may *decrease* social welfare. For example, consider the case when $b_{max} = 2, n_{max} = 2, r = 1$ (shown in blue in Table 1, right) where the payoffs are $(8.0, 2.0)$. This achieves the maximum attainable social welfare by smuggling $n_{max} = 2$ items and having the Sheriff accept a bribe of 2. When $n_{max}$ is increased to 5 (red entry in the table), the payoffs to *both* players drop significantly, and even more so when $n_{max}$ increases further. While counter-intuitive, this behavior is consistent in that the Smuggler may not benefit from loading 3 items every time he was recommended to load 2; the Sheriff reacts by inspecting more, leading to lower payoffs for both players.

That behavior is avoided by increasing the number of rounds $r$: by increasing to $r = 2$ (entry shown in purple), the behavior disappears and we revert to achieving a social welfare of 10 just like in the instance with $n_{max} = 2, r = 1$. With sufficient bargaining steps, the Smuggler, with the aid of the mediator, is able to convince the Sheriff that they have complied with the recommendation by the mediator. This is because the mediator spends the first

| $n_{max}$ | $r = 1$ | $r = 2$ | $r = 3$ |
|---|---|---|---|
| 1 | (3.00, 2.00) | (3.00, 2.00) | (3.00, 2.00) |
| 2 | **(8.00, 2.00)** | (8.00, 2.00) | (8.00, 2.00) |
| 5 | **(2.28, 1.26)** | **(8.00, 2.00)** | (8.00, 2.00) |
| 10 | (1.76, 0.93) | (7.26, 1.82) | (8.00, 2.00) |

Table 1: Payoffs for (Smuggler, Sheriff) in the SW-maximizing EFCE.

$r - 1$ bribes to give a 'passcode' to the Smuggler so that the Sheriff can verify compliance—if an 'unexpected' bribe is suggested, then the Smuggler must have deviated, and the Sheriff will inspect the cargo as punishment. With more rounds, it is less likely that the Smuggler will guess the correct passcode. See also Appendix F in the full paper for additional insights.

## 6 Experimental Evaluation

Even our proof-of-concept algorithm based on the BSSP formulation and subgradient descent, introduced in Section 3, is able to beat LP-based approaches using the commercial solver Gurobi [Gurobi Optimization, 2018] in large games. This confirms known results about the scalability of methods for computing NE, where in the recent years first-order methods have affirmed themselves as the only algorithms that are able to handle large games.

We experimented on *Battleship* over a range of parameters while fixing $\gamma = 2$. All experiments were run on a machine with 64 cores and 500GB of memory. For our method, we tuned step sizes based on multiples of 10. In Table 2, we report execution times when all constraints (feasibility and deviation) are violated by no greater than $10^{-1}$, $10^{-2}$ and $10^{-3}$. Our method outperforms the LP-based approach for larger games. However, while we outperform the LP-based approach for accuracies up to $10^{-3}$, Gurobi spends most of its time reordering variables and preprocessing and its solution converges faster for higher levels of precision; this is expected of a gradient-based method like ours. On very large games with more than 100 million variables, both our method and Gurobi fail—in Gurobi's case, it was due to a lack of memory while in our case, each iteration required nearly an hour which was prohibitive. The main bottleneck in our method was the projection onto $\mathcal{X}_1$ and $\mathcal{X}_2$. We also experimented on the Sheriff game and obtained similar findings (Appendix H in the full paper).

| $(H, W)$ | $r$ | Ship length | #Actions Pl 1 | Pl 2 | #Relevant seq. pairs | Time (LP) $10^{-1}$ | $10^{-2}$ | $10^{-3}$ | Time (ours) $10^{-1}$ | $10^{-2}$ | $10^{-3}$ |
|---|---|---|---|---|---|---|---|---|---|---|---|
| (2, 2) | 3 | 1 | 741 | 917 | 35241 | 2s | 2s | 2s | 1s | 2s | 3s |
| (3, 2) | 3 | 1 | 15k | 47k | 3.89M | 3m 6s | 3m 17s | 3m 24s | 8s | 34s | 52s |
| (3, 2) | 4 | 1 | 145k | 306k | 26.4M | 42m 39s | 42m 44s | 43m | 2m 48s | 14m 1s | 23m 24s |
| (3, 2) | 4 | 2 | 970k | 2.27M | 111M | — out of memory[†] — | | | — did not achieve [‡] — | | |

Table 2: #Relevant seq. pairs is the dimension of $\xi$ under the compact representation of von Stengel and Forges [2008]. For LPs, we report the fastest of Barrier, Primal and Dual Simplex, and 3 different formulations (Appendix G in the full paper). [†] Gurobi went out of memory and was killed by the system after $\sim 3000$ seconds [‡] Our method requires 1 hour per iteration and did not achieve the required accuracy after 6 hours.

## 7 Conclusions

In this paper, we proposed two parameterized benchmark games in which EFCE exhibits interesting behaviors. We analyzed those behaviors both qualitatively and quantitatively, and isolated two ways through which a mediator is able to compel the agents to follow the recommendations. We also provided an alternative saddle-point formulation of EFCE and demonstrated its merit with a simple subgradient method which outperforms standard LP based methods.

We hope that our analysis will bring attention to some of the computational and practical uses of EFCE, and that our benchmark games will be useful for evaluating future algorithms for computing EFCE in large games.

## Acknowledgments

This material is based on work supported by the National Science Foundation under grants IIS-1718457, IIS-1617590, and CCF-1733556, and the ARO under award W911NF-17-1-0082. Gabriele Farina is supported by a Facebook fellowship. Co-authors Ling and Fang are supported in part by a research grant from Lockheed Martin.

## Footnotes

[2]https://github.com/Sandholm-Lab/game-generators    https://github.com/Sandholm-Lab/efce-subgradient

[3]Other CE-related solution concepts in sequential games include the agent-form correlated equilibrium (AFCE), where agents continue to receive recommendations even upon defection, and normal-form coarse CE (NFCCE). NFCCE does not allow for defections during the game, in fact, before the game starts, players must decide to commit to following *all* recommendations upfront (before receiving them), or elect to receive none.

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
