[Reviews · NeurIPS 2019]

Reviewer 1



Originality: To our knowledge, the idea of reformulation of the problem and the proposed benchmarks is new. Quality: While a contribution is the speedup, there is no result about the time complexity. Clarity: The paper is well-organized and easy to understand the problem setting, but the descriptions of the reformulation and the proposed algorithm are not so easy to read. Significant: While the problem setting is for a limited setting, two-player and no chance move, the proposed algorithm and benchmarks are important. The results, however, seem to be suitable for the game theory community rather than the machine learning community. In what follows, I have the detailed comments w.r.t the significant, the quality, and the clarity: -My main concern is that the machine learning community may not be interested in the work. The proposed benchmarks are interesting for the game theory community. The author should describe the reasons why finding an EFCE is interesting for the machine learning community, or proposes other benchmarks that is suitable for machine learning tasks. -The main difference of the proposed algorithm from the previous work [Von Stengel and Forges, 2008] is the efficiency. Thus, the author should describe about the time complexity of the proposed algorithm. -Is there any theorem about the convergence property or the convergence rate of the proposed algorithm? -The authors should denote the setting of two-player and no chance move in the title. -The readers may easily understand the proposed algorithm if pseudocodes are given.

Reviewer 2



This was an overall nice and fun paper to read. The problem was well-motivated and the background well-covered. Possibly some important definitions were skipped (such as sequence form) but with space constraints I feel the authors chose the right level of abstraction for presentation (leaving the right amount of detail for the appendix). One (fairly minor) concern is that the scope is a bit narrow. There is a small community working on these notions of equilibria, and addressing only the 2-player case without chance feels like a bit restrictive, given previous simple algorithms that solve the n-player setting such as Dudik & Gordon. Another concern is that the algorithm is fairly complicated, more so than von Stengel and Forges, which is already fairly complex. Implementing this may be quite involved. If accepted, I would encourage the authors to publish source code; this may help promote the adoption of the benchmarks as well. The benchmark games are indeed interesting. I wonder if there might be any n-player variants where an EFCE would demonstrate a similar pattern of sequential codes (even if they cannot be computed by this algorithm). Any thoughts? Another few suggestions, questions, and comments: - "Contributions: Our primary objective with this paper is to spark more interest in the community towards a deeper understanding of the behavioral and computational aspects of EFCE." This was a refreshing honest statement of the goals, thank you! It set a nice tone for the paper, which I felt delivered on this front. - Is it possible to use a form of online convex optimization, as proposed in Zinkevich 2003 or one of the many in Hazan '16? I seem to recall several methods for projecting back into the feasible sets (ZInkevich uses L2 as well, so possibly the greedy algorithm or its adversarial variant GIGA could be applied here?) - Along similar lines, there was work done on early Poker AI from Gilpin and Hoda and later Kroer, that proposed using Nesterov's excessive gap for two-player zero-sum games (a similar saddle-point formulation). In particular, they develop "treeplex" constraint sets (similar in structure to \chi_1 and \chi_2) and define prox functions that enable the smoothened gradient descent to be efficiently expressed. I wonder if these ideas could now be carried over to solve EFCEs given the reformulation. - I wonder if it another possibility is to use iterative learning, such as fictitious play (which has efficient sequential variants now, see Fictitious Self-Player of Heinrich, Lanctot and Silver '15 and Heinrich & Silver '16, which essentially uses the sequence-form to compute the updates correctly and compactly) - Finally, if approximate solutions are acceptable then it seems another potential route is open up that is even simpler: end-to-end learning with neural networks. The feasibility constraints could be softly implemented by additional L2 loss on the optimization criteria. Might guarantees still be possible with linear models? Even if not, the simplicity of this is very attractive, and there has been a lot of new ideas for training in this saddle point optimization from the adversarial training and generative models community over the last few years.

Reviewer 3



***Originality and Significance*** It is the first time I heard of the concept of EFCE, and it strikes me as a reasonable solution concept which deserves more study. The proposed formulation as bilinear saddle-point problem should be a novel contribution, and even without the experimental simulations I can easily convince myself that it will lead to faster computations of EFCE. I think this contribution itself is sufficient to get the paper accepted in NeurIPS. ***Clarity*** The paper is nicely written. ***Other Major Comments*** As a theoretical researcher, I think the authors should provide a comparison between the LP formulation and the saddle-point formulation. For instance, I would love to see the LP formulation of von Stengel and Forges presented here, so that I can see the structure (e.g., number of variables and constraints; sparse or not) of the LP, which will provide readers more intuition why the saddle-point formulation is more efficient in practice. Also, a very natural question that the author needs to clarify is: does the LP formulation of von Stengel and Forges permit a purely-mathematical reduction to the saddle-point formulation? I ask this because it is known that Nash equilibrium of two-person zero-sum game permits the von-Neumann LP formulation and a saddle-point formulation, for which the LP duality theory establishes the linkage between them. It appears plausible to me this analog to zero-sum game might hint at something more interesting. I am a bit skeptical about the proposed benchmark games. As we see in the experimental results, the run-time becomes very long even when the Battleship game size is raised slightly (and still nowhere close to the real Battleship game we use to play), suggesting the possibility that this benchmark game can hardly be scalable computationally. ====== *** Comments after Rebuttal *** The discussion about (conversion between) LP and saddle-point formulation sounds more interesting to me than the benchmark games and experiments (which took up 3 pages out of 8), so I suggest moving Appendix A to the main paper --- and if the paper is accepted, there will be an extra page available, so no sacrifice on other materials will be needed. I understood that the game states and actions increase rapidly (probably super-exponentially?) in the benchmark games of Battleship, and this is exactly the (more theoretical) reason why I think the benchmark game can hardly be scalable computationally. Abstraction and approximation techniques coming into play might be plausible; since I am not expert on these techniques, I can't gauge how likely these improvements will happen in the future.

[Author Response · NeurIPS 2019]

We thank all reviewers for their nice reviews.

**Reviewer #1:** **[Re "...important results...game theory community rather than the machine learning community"**
**& "the machine learning community may not be interested"]** We are glad to see that the reviewer appreciates the
results in the paper and thinks that they are important. We strongly disagree that this paper is not a good fit for NeurIPS,
and apparently so do the other reviewers. Papers in game theory are explicitly welcomed by NeurIPS (see CFP, which
contains a list of welcomed topics). Also, in recent years there has been growing synergy between the game theory
community and the broader machine learning community. NeurIPS has been an important cog in this process, with
several seminal papers on computational equilibrium finding (e.g., [Zinkevich et al. NeurIPS06] on counterfactual
regret minimization and [Brown&Sandholm NeurIPS17] on the inner workings of the Libratus poker AI). In (at least)
2017 and 2015, NeurIPS best paper awards were given to papers in computational equilibrium finding! **[Re "main**
**difference...from the previous work..."]** This is not accurate: the proposed algorithm uses a first-order method (unlike
the linear programming-based method of von Stengel and Forges), and it is based on a very different formulation
(saddle-point problem instead of linear program). It is a proof of concept of the benefits of the saddle-point formulation:
please see also the great insights of Reviewer #2 on this, which further crystallize the importance of such a formulation.
In any case, the first goal of this paper is *not* to propose a faster algorithm for computing a social-welfare maximizing
EFCE. The main goals and contributions of the paper are as in lines 51–75 in the paper. **[Re "convergence rate of**
**the proposed algorithm?"]** The convergence rate of the algorithm by Wang and Bertsekas is still an open problem.
However, convergence is guaranteed by a supermartingale convergence argument. We'll include a discussion of what is
known about the algorithm's convergence rate in the final version of the paper.

**Reviewer #2:** **[Re "...2-player case without chance...bit restrictive."]** The reason why we focus on the two-player
case with no chance is because in this case a *social-welfare-maximizing* EFCE can be computed in polynomial time,
unlike games with more than two players and/or chance nodes [von Stengel and Forges, 2008]. For the same reason, the
algorithm by Dudik and Gordon does not give any polynomial run-time guarantees about social-welfare-maximizing
EFCE in those more general games. Furthermore, their algorithm operates with normal-form correlation plans—an
exponentially big set—and uses MCMC with a tacit assumption that sampling from the proposed correlated distribution
is practical. Finally, our formulation allows for the computation of EFCEs with *convex* utilities, while their method only
allows for linear (regularized) ones. We'll include this discussion in the final version of the paper. **[Re "...n-player**
**variants..."]** Interesting question! We don't know. Unfortunately, computing social-welfare-maximizing EFCE in
multiplayer games is a hard problem, so this does not seem like an easy task. **[Re "I would encourage the authors to**
**publish source code..."]** Yes! That was already our intention (see Line 63 in our paper), and we agree that it will be a
step in the direction of wider accessibility of the EFCE solution concept. **[Re "...online convex optimization..." &**
**"...Nesterov's excessive gap..."]** Yes! Regret-based methods, as well as Nesterov's EGT algorithm, are techniques that
can solve convex-concave saddle-point problems like EFCE. In a way, that was one of the major inspirations of our
paper: we believe that the saddle-point formulation will be important for designing scalable first-order methods, just
like the saddle-point formulation for Nash equilibrium was crucial for regret-based methods like CFR and EGT-based
methods like the one by Kroer et al. that the reviewer mentioned. The hurdle for designing efficient regret-based
methods is constructing specialized regret minimizers for the polytope of correlated strategies, and the hurdle for EGT
would be designing a smoothing scheme that can be optimized over efficiently (as the reviewer suggests). We hope that
our paper will serve as a starting point for all these interesting directions of exploration. We'll make sure to include this
discussion in the final version of the paper. **[Re "...fictitious play..."]** Yes, fictitious play could be used in this case.
In order to make that efficient, one would need to figure out a way to compute a best response over the polytope of
correlated strategies. **[Re "end-to-end learning"]** Yes! This could be possible by specifying deviators (Line 146) and
correlation plans implicitly as neural networks. We thank the reviewer again for all these incredibly stimulating ideas.

**Reviewer #3:** **[Re "...comparison...LP formulation...saddle-point..." & "...purely-mathematical reduction..."]**
Going from a bilinear saddle-point formulation (i.e., a min-max problem) to a linear program is always possible; in our
case, we show that our saddle-point formulation can be used to recover the LP proposed by von Stengel and Forges (see
Appendix A). The opposite direction is significantly harder: given a minimization problem over a set of variables, it is
not easy to figure out which of those come from the dualization of an internal max problem. Black-box approaches
such as Lagrangian or Fenchel duality are not useful in this case, as they do not recover the original min-max structure:
they simply add more "dual" variables, whose intuitive meaning for the problem at hand is often not immediate.
**[Re "...provide readers more intuition why the saddle-point formulation is more efficient in practice."]** Please
see the answer to Reviewer #2 re "...online convex optimization..." and "...Nesterov's excessive gap...". **[Re "...this**
**benchmark game can hardly be scalable computationally."]** We understand where this comment is coming from.
However, despite their seemingly small size, our game instances have a huge number of states and actions per player
(up to millions). Going forward, an interesting challenge will be around employing abstraction and approximation
techniques that will allow one to scale to larger games and construct mediators that can handle larger interactions.
However, our paper already significantly pushes the boundary of what can be done and has been explored so far by
orders of magnitude. We will discuss this point more in the final version of the paper.

[Meta-Review · NeurIPS 2019]

Game theory is explicitly stated in the CFP of NeurIPS and the scope of the paper, tough narrow, can still interest part of the community. This is the summary of the discussion, as it was the main - if not only - concern about that paper. Reviewers also had different questions/remarks that are integrating in the revised version. I also believe that a journal version of that paper should be submitted somewhere else to have a bigger impact in the game theory and optimization communities.